# CoBEVT: Cooperative Bird's Eye View Semantic Segmentation with Sparse Transformers

**Runsheng Xu**[1]*, **Zhengzhong Tu**[2]*, **Hao Xiang**[1], **Wei Shao**[3], **Bolei Zhou**[1], **Jiaqi Ma**[1]†
[1] University of California, Los Angeles, [2] University of Texas at Austin
[3] University of California, Davis

**Abstract:** Bird's eye view (BEV) semantic segmentation plays a crucial role in spatial sensing for autonomous driving. Although recent literature has made significant progress on BEV map understanding, they are all based on single-agent camera-based systems. These solutions sometimes have difficulty handling occlusions or detecting distant objects in complex traffic scenes. Vehicle-to-Vehicle (V2V) communication technologies have enabled autonomous vehicles to share sensing information, dramatically improving the perception performance and range compared to single-agent systems. In this paper, we propose CoBEVT, the first generic multi-agent multi-camera perception framework that can cooperatively generate BEV map predictions. To efficiently fuse camera features from multi-view and multi-agent data in an underlying Transformer architecture, we design a fused axial attention module (FAX), which captures sparsely local and global spatial interactions across views and agents. The extensive experiments on the V2V perception dataset, OPV2V, demonstrate that CoBEVT achieves state-of-the-art performance for cooperative BEV semantic segmentation. Moreover, CoBEVT is shown to be generalizable to other tasks, including 1) BEV segmentation with single-agent multi-camera and 2) 3D object detection with multi-agent LiDAR systems, achieving state-of-the-art performance with real-time inference speed. The code is available at https://github.com/DerrickXuNu/CoBEVT.

**Keywords:** Autonomous driving, BEV map understanding, Vehicle-to-Vehicle (V2V) application

## 1  Introduction

Autonomous vehicles (AVs) need accurate surrounding perception and robust online mapping capabilities for robust and safe autonomy [1]. AVs are normally located on the ground plane, so it is natural to represent semantic and geometric information of surroundings in the bird's eye view (BEV) maps. Projecting multi-camera views onto the holistic BEV space brings clear strengths in preserving the location and scale of road elements both spatially and temporally, which is critical for various autonomous driving tasks, including scene understanding and planning [2, 3]. It also presents a scalable vision-based solution for real-world deployment without relying on costly LiDAR sensors.

Map-view (or BEV) semantic segmentation is a fundamental task that aims to predict road segments from single- or multi-calibrated camera inputs. Significant efforts have been made toward precise camera-based BEV semantic segmentation. One of the most popular techniques is to leverage depth information to infer the correspondences between camera views and the canonical maps [4, 5, 6, 7]. Another family of works directly learns the camera-to-BEV space transformation, either implicitly or explicitly, using attention-based models [3, 8, 9, 10]. Despite the promising results, vision-based perception systems have intrinsic limitations – camera sensors are known to be sensitive to object occlusions and limited depth-of-field, which can lead to inferior performance in areas that are heavily occluded or far from the camera lens [3, 11].

Recent advancements in Vehicle-to-Vehicle (V2V) communication technologies have made it possible to overcome the limitations of single-agent line-of-sight sensing. That is, multiple connected

---

*Equal contribution. †Corresponding author: jiaqima@ucla.edu.

6th Conference on Robot Learning (CoRL 2022), Auckland, New Zealand.

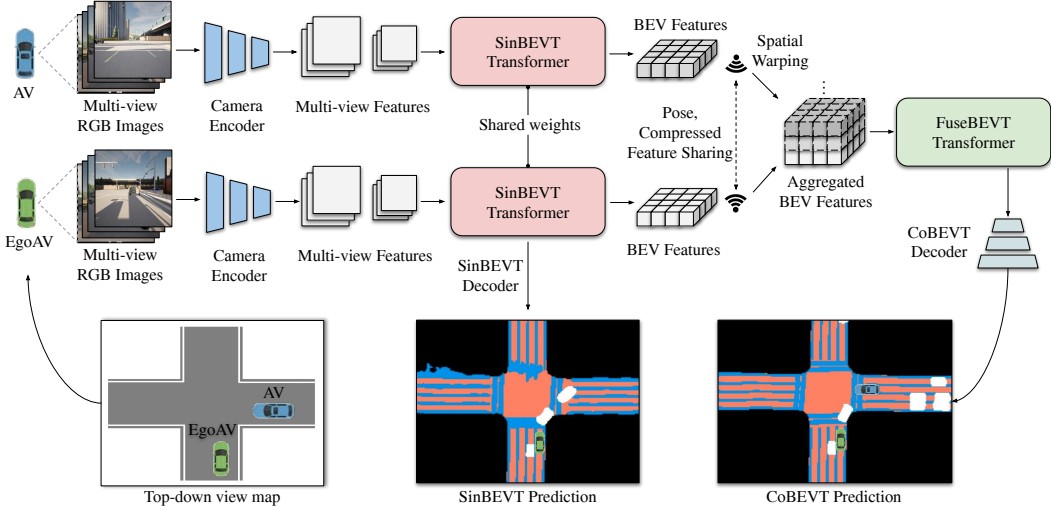

Figure 1: The overall framework of CoBEVT.

AVs can share their sensory information with each other through broadcasting, thereby providing multiple viewpoints of the same scene. Several prior works have demonstrated the efficacy of cooperative perception utilizing LiDAR sensors [12, 13, 14, 15]. Nevertheless, whether, when, and how this V2V cooperation can benefit camera-based perception systems has not been explored yet.

In this paper, we present CoBEVT, the first-of-its-kind framework that employs multi-agent multi-camera sensors to generate BEV segmented maps via sparse vision transformers cooperatively. Fig. 1 illustrates the proposed framework. Each AV computes its own BEV representation from its camera rigs with the SinBEVT Transformer and then transmits it to others after compression. The receiver (i.e. other AVs) transforms the received BEV features onto its coordinate system, and employs the proposed FuseBEVT for BEV-level aggregation. The core ingredient of these two transformers is a novel fused axial attention (FAX) module, which can search over the whole BEV or camera image space across all agents or camera views via local and global spatial sparsity. FAX contains global attention to model long-distance dependencies, and local attention to aggregate regional detailed features, with low computational complexity. Our extensive experiments on the V2V perception dataset [12] show that CoBEVT achieves performance gains of 22.7% and 6.9% over single-agent baseline and leading multi-agent fusion models, respectively.

Furthermore, we demonstrate the generalizability of the proposed framework in two additional tasks. First, we evaluate SinBEVT alone for single-agent multi-view BEV segmentation. Second, we validate the attention fusion on a different sensor modality – multi-agent LiDAR fusion. Our experiments on the nuScenes dataset [16] and the LiDAR-track of OPV2V [12] show that CoBEVT exhibits outstanding performance and capably generalize to many other tasks. Our contributions are:

- We present the generic Transformer framework (CoBEVT) for cooperative camera-based BEV semantic segmentation. CoBEVT delivers superior performance and flexibility, achieving state-of-the-art results on multi-agent camera-based, single-vehicle multi-view BEV semantic segmentation, and multi-agent LiDAR-based 3D detection.

- We propose a novel sparse attention module called fused axial (FAX) attention, which can efficiently capture both local and global relationships between different agents or cameras. We build two instantiations – self-attention (FAX-SA) and cross-attention (FAX-CA) to accommodate different application scenarios.

- We construct a large-scale benchmark study on the cooperative BEV map segmentation task with a total of eight strong baseline models. Extensive experimental results and ablation studies show the strong performance and efficiency of the proposed model. All code, baselines, and pre-trained models will be released.

## 2   Related Work

### 2.1   V2V Perception

V2V perception leverages communication technologies to enable AVs to share their sensing information to enhance the perception. Previous works mainly focus on cooperative 3D object detection with LiDAR. A straightforward sharing strategy is to transmit raw point cloud (i.e. early fusion) [17, 18] or detection outputs (i.e. late fusion) [19]. However, they either require a large bandwidth or ignore the context information. Recently, V2VNet [20] proposes to circulate the intermediate features extracted from 3D backbones (i.e. intermediate fusion), then utilize a spatial-aware graph neural network for multi-agent feature aggregation. Following a similar transmission paradigm, OPV2V [12] employs a simple agent-wise single-head attention to fuse all features. F-Cooper [21] uses a simple `maxout` operation to fuse features. DiscoNet [14] explores knowledge distillation by constraining intermediate feature maps to match the correspondences in the early-fusion teacher model.

### 2.2   BEV Semantic Segmentation

BEV semantic segmentation aims to take camera views as input and predict a rasterized map with surrounding semantics under the BEV view. A common approach for this task is to use inverse perspective mapping (IPM) [22] to learn the homography matrix for view transformation [23, 24, 25]. As camera images lack explicit 3D information, another family of models includes depth estimation to inject auxiliary 3D information [4, 5, 2, 26]. Recently, researchers start to directly model the image-to-map correspondence using transformers or MLPs. VPN [27] learns map-view transformation in a spatial MLP module on flattened camera-view image features. CVT [3] develops positional embedding for each individual camera depending on their intrinsic and extrinsic calibrations. BEV-Former [8] exploits the camera intrinsic and extrinsic explicitly to compute the spatial features in the regions of interest of the BEV grid across camera views using deformable transformer [28].

### 2.3   Transformers in Vision

Transformers are originally proposed for natural language processing [29]. ViT [30] has demonstrated for the first time that, a pure Transformer that simply regards image patches as visual words, is sufficient for vision tasks by large-scale pre-training. Swin Transformer [31] further improves the generality and flexibility of pure Transformers via restricting attention fields in local (shifted) windows. For high dimensional data, video Swin Transformer [32] extends the Swin approach onto shifted 3D space-time windows, achieving high performance with low complexity. Recent works have been focused on improving the architectures of attention models, including sparse attention [33, 34, 35, 36, 37, 38, 39], enlarged receptive fields [40, 41], pyramidal designs [42, 43, 44], efficient alternatives [45, 46, 47], etc.

## 3   Methodology

We consider a V2V communication system where all AVs can exchange sensing information with others. Assuming the poses of all the agents are accurate and transmitted messages are synchronized, we propose a robust cooperative framework that can exploit the shared information across multiple agents to obtain a holistic BEV segmentation map. The overall architecture of CoBEVT is illustrated in Fig. 1, which consists of: SinBEVT for BEV feature computation (Sec. 3.2), feature compression and sharing (Sec. 3.3), and FuseBEVT for multi-agent BEV fusion (Sec. 3.3). We propose a novel 3D attention mechanism called fused axial attention (FAX, Sec. 3.1) as the core component of SinBEVT and FuseBEVT that can efficiently aggregate features across agents or camera views both locally and globally. We will later show that this FAX attention has great generality, showing efficacy on different modalities for multiple perception tasks, including cooperative/single-agent BEV segmentation based on multi-view cameras and cooperative 3D LiDAR object detection.

### 3.1   Fused Axial Attention (FAX)

Fusing BEV features from multiple agents requires both local and global interactions across all agents' spatial positions. On the one hand, neighboring AVs often have different occlusion levels

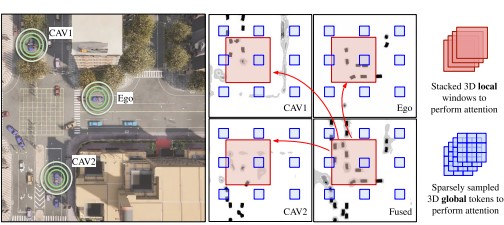 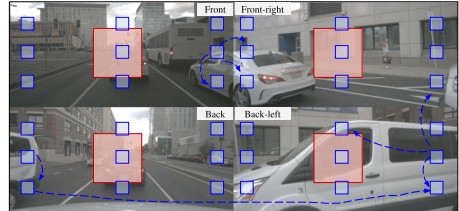

(a) 3D FAX attention for multi-agent BEV fusion.          (b) 3D FAX attention for multi-view fusion.

Figure 2: **Illustrated examples of fused axial attention (FAX) in two use cases – (a) multi-agent BEV fusion and (b) multi-view camera fusion**. FAX attends to both 3D local windows (red) and sparse global tokens (blue) to attain location-wise and contextual-aware aggregation. In (b), for example, the white van is torn apart in three views (front-right, back, and back-left), our sparse global attention can capture long-distance relationships across parts in different views to attain global contextual understanding.

on the same object; hence, local attention, which cares more about details, can help construct pixel-to-pixel correspondence on that object. Take the scene in Fig. 2(a) as an example. The ego vehicle should aggregate all the BEV features per location from nearby AVs to obtain reliable estimates. On the other hand, long-term global contextual awareness can also assist in understanding the road topological semantics or traffic states – the road topology and traffic density ahead of the vehicle are often highly correlated with the one behind. This global reasoning is also beneficial for multi-camera views understanding. In Fig. 2(b), for instance, the same vehicle is torn apart into multi-views, and global attention is highly capable of connecting them for semantic reasoning.

To attain such local-global properties efficiently, we propose a sparse 3D attention model called fused axial attention (FAX), which performs both local window-based attention and sparse global interactions, inspired by [32, 48, 49]. Formally, let $X \in \mathbb{R}^{N \times H \times W \times C}$ be the stacked BEV features with spatial dimension $H \times W$ from $N$ agents. In the local branch, we partition the feature map into 3D non-overlapping windows, each of size $N \times P \times P$. The partitioned tensor of shape ($\frac{H}{P} \times \frac{W}{P}, \underline{N \times P^2}, C$) is then fed into the self-attention model, representing mixing information along the second axis i.e., within local 3D windows [32]. Likewise, in the global branch, feature $X$ is divided using a uniform 3D grid $N \times G \times G$ into the shape ($\underline{N \times G^2}, \frac{H}{G} \times \frac{W}{G}, C$). Employing attention on the first axis of this tensor representing attending to sparsely sampled tokens [48, 49]. Fig. 2 illustrates the attended regions using red and blue colored boxes for local and global branches, respectively.

Combining this 3D local and global attention with typical designs of Transformers [30, 31, 32], including Layer Normalization (LN) [50], MLPs [30], and skip-connections, forms our proposed FAX attention block, as shown in Fig. 3b. Our 3D FAX attention only requires $\mathcal{O}(2(NP)^2 HWC)$ complexity assuming $P \sim G$ (typically $N <= 5$, $P, G \in \{8, 16\}$), significantly cheaper than the full attention $\mathcal{O}((NHW)^2 C)$. Still, it enjoys non-local 3D interactions by seeing through all the agents, which is more expressive than local attention approaches [31, 32]. The 3D FAX self-attention (FAX-SA) block can be expressed as:

$$\hat{\mathbf{z}}^\ell = \text{3DL-Attn}(\text{LN}(\mathbf{z}^{\ell-1})) + \mathbf{z}^{\ell-1}, \qquad \mathbf{z}^\ell = \text{MLP}(\text{LN}(\hat{\mathbf{z}}^\ell)) + \hat{\mathbf{z}}^\ell, \qquad (1)$$

$$\hat{\mathbf{z}}^{\ell+1} = \text{3DG-Attn}(\text{LN}(\mathbf{z}^\ell)) + \mathbf{z}^\ell, \qquad \mathbf{z}^{\ell+1} = \text{MLP}(\text{LN}(\hat{\mathbf{z}}^\ell)) + \hat{\mathbf{z}}^\ell, \qquad (2)$$

where $\hat{\mathbf{z}}^\ell$ and $\mathbf{z}^\ell$ denote the output features of the 3DL(G)-Attn module and MLP module for block $\ell$. The 3DL-Attn and 3DG-Attn represent the above-defined 3D local and global attention, respectively.

### 3.2 SinBEVT for Single-agent BEV Feature Computation

Given monocular views from $m$ cameras on the $i$-th agent $(I_k^i, K_k^i, R_k^i, t_k^i)_{k=1}^m$ denoting input images $I_k \in \mathbb{R}^{h \times w \times 3}$, camera intrinsic $K_k \in \mathbb{R}^{3 \times 3}$, rotation extrinsic $R_k \in \mathbb{R}^{3 \times 3}$, and translation $t_k \in \mathbb{R}^3$, every agent needs to compute a BEV feature representation $\mathbf{F}_i \in \mathbb{R}^{H \times W \times C}$ (height $H$, width $W$, and channels $C$) before any cross-agent collaboration. $\mathbf{F}_i$ can be either fed into a decoder to perform single-agent predictions or shared to the ego vehicle for multi-agent feature fusion.

We take a BEV processing architecture similar to CVT [3], wherein a learnable BEV embedding is initialized as the query to interact with encoded multi-view camera features, as shown in Fig. 3a. We

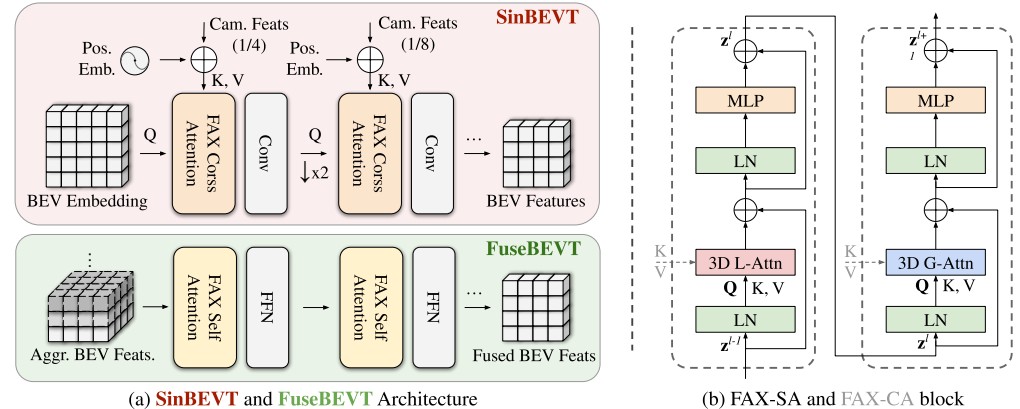

(a) **SinBEVT** and **FuseBEVT** Architecture

(b) FAX-SA and FAX-CA block

Figure 3: Architectures of (a) SinBEVT and FuseBEVT, and (b) the FAX-SA and FAX-CA block.

have observed that CVT uses a low-resolution BEV query that fully cross-attends to image features, which leads to degraded performance on small objects, despite being efficient. Thus, CoBEVT learns a high-resolution BEV embedding instead, then uses a hierarchical structure to refine the BEV features with reduced resolution. To efficiently query features from camera encoders at high resolution, the FAX-SA module is further extended to build a FAX cross-attention (FAX-CA) module (Fig. 3b), in which the query vector is obtained using the BEV embedding, whereas the key/value vectors are projected by multi-view camera features. Before applying cross-attention, we add a camera-aware positional encoding derived from camera intrinsics and extrinsics, to learn implicit geometric reasoning from individual camera views to a canonical map-view representation, following CVT. This rather simple, implicit approach demonstrates a good balance of performance and efficiency, and our FAX attention allows for global interactions in a hierarchical network, showing better accuracy against low-resolution isotropic approaches such as CVT.

### 3.3 FuseBEVT for Multi-agent BEV Feature Fusion

**Feature Compression and Sharing.** Transmission data size is critical to V2V applications, as large bandwidth requirements will likely cause severe communication delays. Therefore, it is necessary to compress the BEV features before broadcasting. Similar to [13, 14], we apply a simple 1x1 convolutional auto-encoder [51] to compress and decompress the BEV features. Once receiving the broadcasted messages that contain intermediate BEV representations, and the pose of the sender, the ego vehicle applies a differentiable spatial transformation operator $\mathbf{\Gamma}_\xi$, to geometrically warp the received features [52] onto the ego's coordinate system: $\mathbf{H}_i = \mathbf{\Gamma}_\xi\left(\mathbf{F}_i\right) \in \mathbb{R}^{H \times W \times C}$.

**Feature Fusion.** We design a customized 3D vision Transformer called FuseBEVT that can attentively fuse information of the received BEV features from multiple agents. The ego vehicle first stacks the received and projected BEV features $\mathbf{H}_i,\ i = 1, ..., N$ into a high dimensional tensor $\mathbf{h} \in \mathbb{R}^{N \times H \times W \times C}$, then feeds them into the FuseBEVT encoder which consists of multiple layers of FAX-SA blocks (Fig. 3a). Benefiting from the linear complexity of FAX attention (Sec. 3.1), this agent-wise fusion Transformer is also efficient. Each FAX-SA block conducts a 3D global and local BEV feature transformation via Eqs. 1-2. As exemplified in Fig. 2(a), the 3D FAX-SA can attend to the same region of estimations (red boxes) drawn from multiple agents to derive the final aggregated representations. Moreover, the sparsely sampled tokens (blue boxes) can interact globally to attain a contextual understanding of the map semantics such as road, traffic, etc.

**Decoder.** We apply a series of lightweight convolutional layers and bi-linear upsampling operations on the aggregated BEV representation and generate the final segmentation output.

## 4   Experiments

We evaluate the effectiveness of the proposed CoBEVT on the camera track of the V2V perception dataset OPV2V [12]. To show the flexibility and generality of our CoBEVT, we also conduct experiments on the LiDAR track of OPV2V and the autonomous driving dataset nuScenes [16].

Table 1: **Map-view segmentation on OPV2V camera-track.** We report IoU for all classes. All fusion methods employs CVT [3] backbone, except for CoBEVT which uses SinBEVT backbone.

| Method | Veh. | Dr.Area | Lane |
|---|---|---|---|
| No Fusion | 37.7 | 57.8 | 43.7 |
| Map Fusion | 45.1 | 60.0 | 44.1 |
| F-Cooper [21] | 52.5 | 60.4 | 46.5 |
| AttFuse [12] | 51.9 | 60.5 | 46.2 |
| V2VNet [20] | 53.5 | 60.2 | 47.5 |
| DiscoNet [14] | 52.9 | 60.7 | 45.8 |
| FuseBEVT | **59.0** | **62.1** | **49.2** |
| CoBEVT | **60.4** | **63.0** | **53.0** |

Table 2: **3D detection results on the OPV2V LiDAR-track.** All methods employ PointPillar [61] backbone. (C) denotes using $64\times$ feature compression.

| Method | AP0.7 | AP0.7(C) |
|---|---|---|
| No Fusion | 60.2 | 60.2 |
| Late Fusion | 78.1 | 78.1 |
| Early Fusion | 80.0 | - |
| F-Cooper | 79.0 | 78.8 |
| AttFuse | 81.5 | 81.0 |
| V2VNet | 82.2 | 81.4 |
| DiscoNet | 83.6 | 83.1 |
| FuseBEVT | **85.2** | **84.9** |

Table 3: **Vehicle map-view segmentation on nuScenes.** All models use only a single time-stamp. $*$ denotes our reproduced result with EfficientNet-b4 backbone.

| Method | Veh. | Par(M) | FPS |
|---|---|---|---|
| VPN* [27] | 29.3 | 4. | 31 |
| OFT [59] | 30.1 | - | - |
| Lift-Splat | 32.1 | 14 | 25 |
| FIERY [4] | 35.8 | 7 | 8 |
| CVT [3] | 36.0 | 1.2 | 35 |
| SinBEVT | **37.1** | 1.6 | 35 |

## 4.1 Datasets and Evaluations

**OPV2V** is a large-scale V2V perception dataset that is collected in CARLA [53] and the cooperative driving automation tool OpenCDA [54]. It contains 73 diverse scenarios, which have an average of 25 seconds duration. In each scenario, various numbers (2 to 7) of AVs show up simultaneously, and each one is equipped with one LiDAR sensor and 4 cameras towards different directions to cover 360° horizontal field-of-view. Our main experiment only utilizes the camera rigs of the dataset, and we use Intersection over Union (IoU) between map prediction and ground truth map-view labels as the performance metric. Since OPV2V has multiple AVs in the same scene, we select a fixed one as the ego vehicle during testing and evaluate 100m×100m area around it with a 39cm map resolution.

To demonstrate its generality, we also evaluated our proposed CoBEVT on the OPV2V LiDAR-track 3D detection task. We use the same evaluation range in [12, 55], and the detection performance is measured by Average Precisions (AP) at an IoU threshold of 0.7. For both camera and LiDAR track, there are 6764/1981/2719 frames for train/validation/test set, respectively.

**The nuScenes dataset** contains 1000 diverse scenes, each of around 20 seconds long. In total, there are 40K sampled frames in this dataset, and the dumped data captures a 360° view of surroundings using 6 cameras. We use the groundtruth in [3]. The evaluation ranges are [-50m, 50m] for the X and Y axis, and the resolution of the BEV grid is 0.5m.

## 4.2 Experiments Setup

**Implementation details.** We assume all the AVs have a 70m communication range following [20], and all the vehicles out of this broadcasting radius of ego vehicle will not have any collaboration. For the OPV2V camera-track,we choose ResNet34 [56] as the image feature extractor in SinBEVT. The transmitted BEV intermediate representation has a resolution of $32 \times 32 \times 128$. For the multi-agent fusion, our FuseBEVT component has 3 encoded layers and a window size of 8 for both local and global attention. We train the whole model end-to-end with Adam [57] optimizer and cosine annealing learning rate scheduler [58]. We use weighted cross entropy loss and train all models with 60 epochs, with a batch size of 1 per GPU. Please refer to the supplementary materials for more details, as well as the configurations on nuScenes and OPV2V LiDAR-track.

**Compared methods.** For multi-agent perception task, we consider single-agent perception system *No Fusion* as the baseline. We compare with the state-of-the-art multi-agent perception algorithms: F-Cooper [21], AttFuse [12], V2VNet [20], and DiscoNet [14]. We also implement a straightforward fusion strategy *Map Fusion*, which transmits the segmentation map instead of BEV features and fuses all maps by selecting the closest agent's prediction for each pixel.

For the nuScenes dataset, we compare against state-of-the-art models including CVT [3], FIERY [4], View Parsing Network (VPN) [27], Orthographic Feature Transform (OFT) [59], and Lift-Splat-Shoot [5]. All models only utilize single-step timestamp data for fair comparisons. We intentionally use the same image feature extractor Efficient-B4 [60] and decoder as CVT and FIERY.

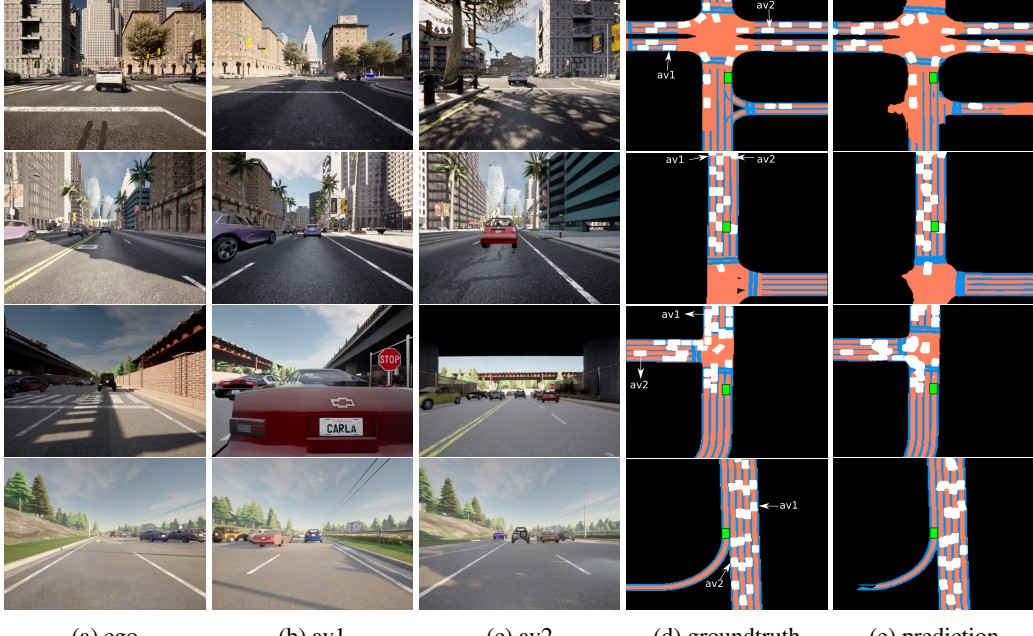

|         (a) ego         |         (b) av1         |         (c) av2         |      (d) groundtruth      |       (e) prediction      |

Figure 4: **Qualitative results of CoBEVT.** From left to right: the front camera image of (a) ego, (b) av1, (c) av2, (d) groundtruth and (e) prediction. The green bounding boxes represent ego vehicle, while the white boxes denote the segmented vehicles. CoBEVT demonstrates robust performance under various traffic situations and road types.

## 4.3    Quantitative Evaluation

**OPV2V camera-track results.** To make a fair comparison, we first employ CVT [3] to extract the BEV feature from camera rigs for all methods and only use the fusion component (i.e. FuseBEVT) of CoBEVT to compare with other fusion models. Then we compare it with our complete CoBEVT to show the effectiveness of SinBEVT as well. As shown in Tab. 1, all cooperative methods perform better than *No Fusion*, which proves the benefits from multi-agent perception system. Among all fusion models, our FuseBEVT achieves the best IoU for all classes, outperforming the second-best method by 5.5%, 1.4%, and 3.4% on vehicle, drivable area, and lane, respectively. More importantly, by replacing the CVT with our SinBEVT for feature extraction, our CoBEVT can further increase the accuracy by 1.4%, 0.9%, and 3.8% on the three classes compared to using FuseBEVT only.

**OPV2V LiDAR-track results.** As Tab. 2 reveals, our FuseBEVT also has the best performance on the LiDAR-track task, which improves the single-agent system by 25.0% and outperforms the leading algorithm DiscoNet by 1.7%. Furthermore, our method exhibits great robustness against LiDAR feature compression, with only a 0.3% drop with the $64\times$ compression rate.

**nuScenes vehicle map-view segmentation.** Our SinBEVT can run 35 FPS on RTX2080 with 37.1 IoU score and 1.6 M parameters, achieving the best accuracy with real-time performance. Compared to the SOTA method CVT, we are 1.1% higher with similar parameters and latency.

**Effect of compression rate.** Data transmission size is a critical factor in V2V applications. Here we study the effect of different compression rates on our CoBEVT by adjusting the $1 \times 1$ convolution. Tab. 4 shows that CoBEVT is insensitive to compression, and it can still beat other fusion methods even with a large compression rate of 64.

Table 4: **Compression effect on OPV2V Camera.**

| CPR-rate | Size (KB) | IoU |
|---|---|---|
| 0x  | 524 | 60.4 |
| 8x  | 66  | 60.1 |
| 16x | 33  | 58.9 |
| 32x | 16  | 56.2 |
| 64x | 8   | 54.8 |

## 4.4    Qualitative Analysis

Fig. 4 shows the qualitative results of CoBEVT on scenes containing 3 AVs. In each row, we draw the front camera image of each AV along with the ground truth and prediction pairs. Our framework

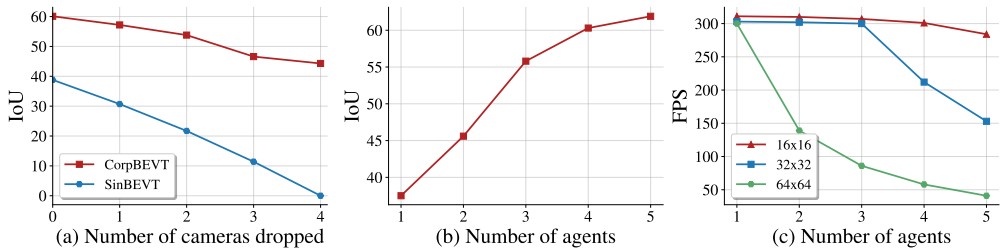

Figure 5: **Ablation studies.** (a) IoU *vs.* number of dropped cameras (b) IoU *vs.* number of agents. (c) FPS *vs.* number of agents. The channel dimension of BEV feature map is fixed as 128 for (c).

can overcome most of the occlusions and perceive distant objects accurately, benefiting from our Transformer design that learns from all agents and views.

### 4.5 Ablation Study

**Component analysis.** Tab. 5 shows the importance of local and global attention. Both attention blocks significantly contribute to the final performance.

**Robustness to camera dropout.** Sensor failure during driving can lead to fatal accidents. Therefore, here we investigate how well our CoBEVT handles it. We random drop $n \in [1, 4]$ cameras of the ego vehicle, and demonstrate the performance decrease for both SinBEVT (no collaboration) and CoBEVT in Fig. 5a. It can be seen that by introducing sensing cooperation, driving safety can be significantly improved, as even if all ego cameras break down, CoBEVT can still reach an IoU score of 44.3.

Table 5: **Component ablation.**

| Local | Global | Veh./Dr.Area/Lane |
|:-----:|:------:|:-----------------:|
|       |        | 52.6/57.9/42.0    |
| ✓     |        | 57.8/61.5/49.2    |
|       | ✓      | 57.9/60.8/48.6    |
| ✓     | ✓      | **60.4/ 63.0 /53.0** |

**Number of agents.** Here we study the influence brought by the number of collaborators on CoBEVT. As Fig. 5b describes, increasing the collaborators can generally bring performance improvement, whereas such gain will be marginal when the agent number is greater than 4.

**Inference speed of FuseBEVT.** Real-time multi-agent feature fusion is critical for real-world deployment. Here we examine the inference speed of FuseBEVT with different BEV feature map spatial resolution (from 16 to 64) and the number of agents on RTX3090. Fig. 5c shows that our fusion algorithm can achieve real-time performance under distinct collaboration scenarios.

## 5 Conclusion and Limitations

In this paper, we propose a holistic vision Transformer dubbed CoBEVT for multi-view cooperative semantic segmentation. We propose a fused axial attention (FAX) mechanism that allows for local and global interactions across all views and agents. Extensive experiments on both simulated and real-world datasets show that CoBEVT achieves superior performance on BEV segmentation.

**Limitations.** The proposed approach does not explicitly model realistic V2V challenges such as asynchronization and position errors, which may impair its robustness under these noises. Addressing these limitations needs future research on real-world, realistic, and diverse cooperative datasets and benchmarks.

**Acknowledgments**

This material is supported in part by the Federal Highway Administration Exploratory Advanced Research (EAR) Program, and by the US National Science Foundation through Grants CMMI # 1901998. We thank Xiaoyu Dong for her insightful discussions.

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
