# OpenReview forum: "CoBEVT: Cooperative Bird’s Eye View Semantic Segmentation with Sparse Transformers"
_robot-learning.org/CoRL/2022/Conference — CoRL 2022 Poster_

### Official Review · Reviewer_gxHq · 2022-07-31

**Originality:** Good
**Technical Quality:** Very Good
**Clarity Of Presentation:** Good
**Impact:** 4

**Recommendation:**

Weak Accept: I recommend accepting the paper, but will not argue for my recommendation if the majority of other reviewers have a different opinion.

**Summary:**

The paper presents a solution to attention-based multi-sensor multi-agent camera data fusion for the task of bird's eye view (BEV) semantic segmentation. The proposed framework CoBEVT depends on a novel fused axial attention (FAX) mechanism at both the feature extraction and fusion phases as part of Transformer architectures. The FAX mechanism conducts attention at both the local and global scales, while remaining tractable. CoBEVT is empirically validated as part of a simulated (in CARLA) vehicle-to-vehicle benchmark and on the real-world NuScenes dataset. CoBEVT outperforms state-of-the-art baselines across cooperative BEV semantic segmentation, singe-agent multi-camera BEV semantic segmentation, and 3D object detection with multi-agent LiDAR sensors tasks with real-time inference speed.

**Issues:**

In addition to the weaknesses listed above, I am hoping the authors can address the following clarifications and discussion points during the rebuttal period.

1. Are the stacked BEV features from $N$ agents (line 130) already transformed into the same spatial area?
2. How could object disappearance and merging of multiple vehicles in the predictions be addressed (e.g., in the appendix Fig. 3, third row example).
3. Are there ways to enforce known road geometry to avoid deformation in the predictions?
4. In Fig. 5, is $32 \times 32$ the spatial resolution used in the earlier experiments?
5. It would be helpful to include the ground truth in the appendix Figs. 4 and 5. What is the difference between CoBEVT and DiscoNet qualitative results in these two figures?

**Quality Of The Limitations Section:**

Limitations are addressed clearly

**Reviewer Expertise:**

3: The reviewer is fairly confident that the evaluation is correct

**Robotics Focus:**

Highly relevant to robotics but no hardware experiments

**Strengths And Weaknesses:**

Strengths:
* The biggest strength of the paper is its empirical validation. The authors do their due diligence in thoughtfully constructing experiments that showcase the advantages of the CoBEVT framework as compared to a significant number of baselines across several tasks. The ablation studies clearly show the benefits of both local and global attention, robustness to sensor failure, multi-agent sensor fusion, and BEV feature map compression on inference speed.
* The camera-based multi-sensor multi-view CoBEVT framework appears to be novel given a quick search of existing literature. The authors do a good job contextualizing their approach in prior work.
* The limitations section is honest and thorough providing helpful context and opportunities for future work.
* Fig. 1 is an effective illustration of the CoBEVT framework.

Weaknesses:

My biggest concerns with the paper center around the presentation and polish.
* In Sec. 2.2 and 2.3 of the literature review, it would be helpful to include a sentence in each section relating how the described literature relates to the presented approach.
* Fig. 2 lacks clarity and polish. It is unclear how the BEV image on the left in Fig. 2(a) corresponds to the predicted scenes on the right. Occluded vehicles should be highlighted in the figure. The captions for the two subfigures should be better spaced out. In Fig. 2(b), the different views (side, back, front) should be labeled. From the caption and the text in the body of the paper, it is not entirely clear what is the takeaway from Fig. 2(b). For example, which car is torn apart into multiple views? Is it the one in the lower row? How does the illustrated attention combat this?
* In methods lines 128-137, it was not really clear the difference between local and global attention. The notation makes it seem like they are the same thing but transposed. Further illustrating the difference between the approaches would be helpful here. Additionally, since the sparsity in the attention is critical to the proposed CoBEVT approach, a description of how sparsity is achieved should be included either in the main paper or in the appendix.
* In methods lines 143-145, it is not clear why the equations for local and global attention have notation that differs by one index value or why they are sequential. More context and explanation is necessary here. The 'LN' acronym is not defined in the paper.
* Table 1, 2, and 3 captions are too close together and have spelling errors (PointPillar, nuScene - both should be plural).
* In Fig. 4, the white boxes (vehicles) were not explained in the caption. The occluded and far-away vehicles should be indicated in the figure to showcase the performance of CoBEVT. The merging of vehicles in the predictions should be listed as a limitation.
* In Table 5, it is unclear what is the structure of the no local and global attention FAX architecture (first table row).

The paper currently contains many typos and requires polishing. The following is a non-exhaustive list of typos as well as grammatical and consistency issues I found:

1. Lines 3-5: incorrect grammar in 'they are all based on single-agent camera-based systems which are difficult to handle occlusions and detect distant objects in complex traffic scenes'.
2. Fig. 1: white blocks are not indicated to be cars.
3. Line 51-52: 'FAX contains global attention to reason high-level contextual information' should read 'about high-level contextual information'.
4. Line 59: 'show that CoBEVT exhibit' should be 'exhibits'.
5. Line 112: 'We propose a novel 3D attention mechanism ... as the core components' should be 'component'.
6. Line 113: there should be no comma before 'that'.
7. Line 115: 'sing-agent' should be 'single-agent'.
8. Line 124: 'ahead the vehicle' should be 'ahead of the vehicle'.
9. Lines 140-143: 'Our 3D FAX ... more expressive than local approaches [27, 28].' sentence is too long and has grammatical errors.
10. Lines 176-177: 'The ego vehicle first stacks the received and projected BEV features ..., then feed into' should be 'then feeds them into'.
11. Line 182: 'sparsely sampled tokens are interacted globally' should be 'tokens interact globally'.
12. Line 192: '4 cameras towards distinction to cover' is incorrect grammar.
13. Line 201: 'the detection performance is measured with Average Precisions (AP) at IoU thresholds 0.7' should be 'Average Precision (AP) at an IoU threshold of 0.7'.
14. Line 206: 'the size of resolutions of the BEV grid is 0.5m' should be 'the resolution of the BEV grid is 0.5 m'.
15. Line 212: 'layer3' should be 'layers'.
16. Line 213: 'Adaw' should be 'Adam'.
17. Line 243: 'stat-of-the-art' should be 'state-of-the-art'.
18. Line 246: Missing space between sentences.
19. Line 247: Missing space between 'the' and '$1 \times 1$'.
20. The references have typos and consistency issues. For example, 'Bev' in [1] should be capitalized. Conference acronyms are sometimes included (e.g., IV in [6]) and sometimes not (e.g., [3]). When possible, the conference venue should be cited instead of ArXiv (e.g., [1]). The references should in general be proofread (e.g., [6] lists the year and IEEE twice).

**Summary Of Recommendation:**

Overall, the paper presents a novel approach to multi-agent multi-sensor BEV semantic segmentation, which is of interest to the autonomous driving community. A new scalable attention mechanism is introduced for the task. The experiments extensively validate the proposed approach. However, the current presentation lacks polish and clarity. As it stands, I am leaning towards rejecting the paper in its current state. I am open to increasing my score based on the rebuttal discussion and updates made to the paper during the rebuttal phase. That being said, I think this is a promising paper, and with some improvements to the presentation it will make a good contribution.

**Post-Rebuttal**: Upon going through the authors' rebuttal, I am inclined to raise my score to a Weak Accept. The authors put in significant effort in terms of clarifications and further exposition to address my concerns. Although I still have some concerns surrounding clarity and polish, I believe that the proposed method is novel and interesting with extensive and promising empirical results.

---

> ### Author Response · Authors · 2022-08-22
> **Response to Reviewer gxHq**
>
>
> We would like to thank the reviewer for the detailed comments and corrections on the paper presentation. We have fixed all the grammar issues spotted by the reviewer and proofread again to improve the clarity of our paper. Below are responses to the additional issues raised by the reviewer.
>
> - **Literature review.** Thank you for the great suggestion. We have added extra sentences to better relate the related work to our proposed approach.
>
> - **Regarding Fig. 2.** We thank the reviewer for bringing up this issue. Firstly, the black color in the prediction image of Fig. 2(a) represents the probability map for vehicle segmentation. To help readers see the correspondence between the CARLA scenario under BEV on the left and the predictions on the right side, we draw the road topology in green overlayed with the predictions, which can help readers find the groundtruth-prediction pair more conveniently.
> Secondly, we have added the takeaway and analysis of Fig. 2(b) in the revised paper. Hope it can help for better understanding.
>
> - **Clarity of the 3D FAX attention (lines 128-137, lines 143-145).**
> We thank the reviewer for the comments on the clarity about FAX attention. We have added an entire Section 2 in the supplementary materials hope of helping better understand.
> The reviewer is correct that these two approaches seem like the same thing but only transposed, especially in terms of implementation.
> The sparsity naturally comes from the local and global attention we design -- instead of attending to all the pixels in the 3D space, our proposed FAX attention first attends to a small 3D window, then a sparsely sampled grid of pixels.
> We have added the definition of "LN" acronym before its use.
>
> - **Tables 1,2,3 and spelling errors.**
> Thanks for the correction. We have made the captions more separated and revised the typos.
>
> - **Table 4.**
> We have added explanations of white boxes in the caption of Fig. 4. We have also added markers on the occluded and far-away vehicles in the figures. The limitations of the "merging predictions" have been added in the qualitative experiments section.
>
> - **Table 5.** We ablate the global and local attention in FuseBEVT to investigate their effectiveness. If both two attention modules are removed (i.e., first row), the FuseBEVT only has two consecutive MLP layers without any spatial mixing.
>
> - **Are the stacked BEV features from $N$ agents (line 130) already transformed into the same spatial area?**
>     Yes, as we mentioned in Section 3.3, after the ego vehicle receives the BEV representations from all CAVs, then applies a differentiable spatial transformation operator $\mathbf{\Gamma_{\xi}}$, to geometrically warp the received features onto the ego's coordinate system:  $\mathbf{H_i}=\mathbf{\Gamma_{\xi}}\left(\mathbf{F}_i\right)\in \mathbb{R}^{H \times W \times C}$.
>
> - **How could object disappearance and merging of multiple vehicles in the predictions be addressed?**
>     The global attention in our proposed FAX module can reason long-term relationships across multiple views, thus capable of capturing global contextual information. Fig. 2 (b) illustrates an example wherein the white van is torn into multiple views, and our global attention (blue boxes) can connect all the parts across views.
>
> - **Are there ways to enforce known road geometry to avoid deformation in the predictions?**
>     A straightforward solution would be to retrieve basic road geometry from OpenStreetMap and input it as the prior knowledge to the network. For example, we can train a separate neural network that takes both initial map prediction and a rasterized map from OpenStreetMap's road topology as input to obtain a refined predicted BEV map.
>
> - **In Fig.5, is $32\times 32$ the spatial resolution used in the earlier experiments?**
>     Yes, in the OPV2V experiments, all the multi-agent algorithms use $32\times32$ spatial resolution for BEV features. The BEV features will be further upsampled to $256\times256$ to get the final BEV map.
>
> - **Difference between CoBEVT and DiscoNet qualitative results in these two figures?**
>     Fig. 4 and 5 in the supplementary materials show that our CoBEVT can produce more accurate bounding box predictions for occluded and distant vehicles. We have highlighted the prediction difference of CoBEVT and DiscoNet using **red circles** in Appendix Figs. 4 and 5, which we hope can better for better visualization.
> - **Typos and grammatical issues.**
>     We sincerely appreciate the reviewer for spotting all these issues.
>     We have fixed them all and proofread them again for better readability.
>
> Best,
>
> Authors of paper 142

---

> > ### Comment · Reviewer_gxHq · 2022-08-25
> > **Response to Authors**
> >
> > Thank you for your response to my concerns above. I list a few follow-ups below.
> >
> > I am not sure my intended meaning for the following question was clear:
> >
> > > How could object disappearance and merging of multiple vehicles in the predictions be addressed?
> >
> > As now listed in the limitations, the predictions for certain vehicles get merged together (losing vehicle individuality) or disappear. Do you have some hypotheses for how this might be avoided within the proposed method in future work?
> >
> > Unfortunately, I am still not very clear on the intuitive difference between local and global attention based on the notation presented in the main paper and in Section 2 of the appendix. For example, in Eq. 3 (appendix), what operations are the arrows doing? The first one is the partitioning operation as far as I understand. Is the second one a neural network or just a transpose? Is the idea here that rearranging the image grid in certain ways allows the attention to be applied to different dimensions? Intuition for what makes Eq. 3 local vs Eq. 4 global attention would be helpful, since this is a critical component of the contribution.
> >
> > Also, please note the 'paritioning' typo throughout this section.
> >
> > The qualitative results in Section 4 are sparse in discussion and description. It is not clear what the reader should be looking for in, for example, Fig. 4 based lines 102-105 and the figure caption. The red circles are helpful but they need to be supplemented with verbal descriptions for what the reader should pay attention to in the qualitative results.
> >
> > In line 103, the reference is currently broken.

---

> > > ### Author Response · Authors · 2022-08-26
> > > **Response to Reviwer gxHq**
> > >
> > > We sincerely thank the reviewer for the follow-up discussions and apologize for any confusion in the last round's response. We have fixed the typo in the supplementary file.
> > > - **How could object disappearance and merging of multiple vehicles in the predictions be addressed:** Thanks for explaining this question again. In future work,  we can add a specific loss called "merging loss" to punish the merging vehicles. For example, we can generate a mask in the tight space between the neighbors for close vehicles. If the network makes positive predictions in the masked area, it will be punished with a higher loss. The vehicles that disappear in the predictions (false negatives) usually are the ones that are highly occluded from all agents' views, which are very challenging to detect. In OPV2V dataset, our implemented loss is cross-entropy loss with fixed weights for certain classes (e.g., background, vehicle). To solve the disappearing issue, we can replace it with focal loss, which is known to handle challenging examples well.
> > >
> > > - **Local and global attention:** Yes, you are right that both arrows in Eqs. 3 and 4 mean "Transpose." Yes, to reuse the same attention, we rearrange the image grid in certain ways to allow attention to be applied on the "(global) grid axis."
> > > We have added a piece of pseudocode, Algo. 1 in the appendix to help understand the details better. Note that in the equations, we changed to use of two different parameters, $P$, and $G$, to avoid confusion. Please also check our uploaded code for the PyTorch implementations in the original submission. We also feel that it is a bit difficult to formulate the proposed FAX attention using equations, so we have added enough visualizations of the FAX attention areas in Figs. 2 and 3 in the main paper for intuitive understanding, should the readers prefer.
> > >
> > > - **Qualitative Results:**  As the reviewer suggested, we have added more descriptions on the OPV2V LiDAR-track qualitative results part. We have added some texts near the red circles in Fig. 4 and Fig. 5 in the supplementary file to make readers easier to understand the visual difference between ours and DiscoNet.

---

> ### Author Response · Authors · 2022-08-22
> **Revised Supplementary Materials**
>
> We have attached the revised supplementary materials to this response. We have highlighted the changes in blue color.

---

### Official Review · Reviewer_KVMN · 2022-08-01

**Originality:** Good
**Technical Quality:** Fair
**Clarity Of Presentation:** Very Good
**Impact:** 3

**Recommendation:**

Weak Reject: I recommend rejecting the paper, but will not argue for my recommendation if the majority of other reviewers have a different opinion.

**Summary:**

The paper proposes a method for RGB-alone BEV perception across multiple agents and views. Each agent uses a perspective-to-BEV attention mechanism to construct a BEV feature map, and  sends its BEV feature maps to other agents. Then, each agent stacks all BEV feature maps and computes a fused BEV feature map.  The paper proposes as its contribution a hierarchical transformer in the BEV space, where attention is first performed within local windows, and then across far away locations on the map. They evaluate their model on both single agent and multi-agent RGB alone BEV segmentation and LiDAR 3D detection.

**Issues:**

Could you use your model architecture and compare FAX with a generic 2D transformer in the BEV space, a SWIN transformer and a 2D CNN with varying number of layers in terms of accuracy and computational cost? Could you repeat these comparisons with double the resolution of the BEV map?


Why building a BEV per agent is important? What if the multiview images across all agents are broadcasted, and a transformer directly attends into them to build the complete BEV map of the scene?

I am happy to update my score if the authors address the questions above.


Post rebuttal: I greatly appreciate the effort  of the authors for providing additional experiments.

The authors argue that https://arxiv.org/pdf/2206.07959.pdf showed up on arxiv post CoRL submission which is correct. I was pointing though  to the Table 2 of that paper, where you can see *earlier* papers, such as  BEVformer having *much* higher IoU in Nuscenes than the proposed model. Essentially this means the present paper is much below the SOTA. (BEVformer was submitted on arxiv on 31 Mar 2022)

Fro the comparison against other transformer variant and number of layers of those that the authors provide,  we see that the proposed model has marginally better performance (2%) with slightly less number of FLOPS . I think 2% difference is small given that changing the resolution had a 7% boost (as they report), and the best proposed method's variant provided at rebuttal time is already 5% below the SOTA in Nuscenes in comparison to BEVformer.

By broadcasting images i meant broadcasting perspective features instead of BEV features, apologies if this was confusing.

Overall, this is a reasonable paper that is simply not moving as fast as the SOTA in the field of  multi-view BEV fusion. I deeply appreciate the effort of the authors, and I think their additional experiments have elucidated what really matters for performance in BEV. For the reasons outlined above, I maintain my original rating.



**Quality Of The Limitations Section:**

Limitations are not well addressed

**Reviewer Expertise:**

4: The reviewer is confident but not absolutely certain that the evaluation is correct

**Robotics Focus:**

Relevant but unlikely to deploy to hardware in near future

**Strengths And Weaknesses:**

+ The paper is overall clearly written and addresses an important problem

-The main contribution of the paper seems to be the sparse attention module (FAX). There are no ablations where everything is kept the same in the model, and instead a generic transformer is used to fuse information in the BEV map, or a SWIN transformer or a 2D CNN. the ablations consider previous fusion mechanisms of other methods (Table 1), and the reader does not know neither what fusion method the previous methods used, nor whether there are more changes between the present model and the models of the previous paper. The proposes sparse  transformer is never evaluated in apples-to-apples comparison with other choices.

- recent papers on arxiv can obtain way higher numbers than the present paper without any transformer  for fusion of information and without even any multi agent collaboration, at least in the tasks of BEV segmentation, e.g., https://arxiv.org/pdf/2206.07959.pdf table 2. This casts some doubt on the validity of the conclusions of the present paper, since hyperparameters such as resolution or batchsize are not investigated, and the reader does not know if the gains survive such hyperparameter changes.

**Summary Of Recommendation:**

The paper addresses an important problem but does not provide convincing ablations of its main contribution, which is the FAX module and its numbers are significantly lower than recent papers in the literature.

---

> ### Author Response · Authors · 2022-08-22
> **Response to Reviewer KVMN**
>
> We appreciate the reviewer for acknowledging the contributions and significance of our work. We provide pointwise responses to your concerns below.
>
> - **Introduction to previous state-of-the-art multi-agent fusion methods.**
>     Previous multi-agent fusion methods mainly focus on the intermediate fusion strategy, where the extracted deep neural features from 3D LiDAR points are shared to perform the fusion. After receiving the BEV features from nearby agents, F-Cooper [1] directly applies an element-wise \textit{maxout} operation on the spatial dimension across different agents. AttFuse [2] constructs a local graph using single-head attention for each spatial location in the feature map, where edges are built for feature vectors in the same spatial location from disparate agents. V2VNet [3] leverages a fully-connected graph neural network (GNN) as the aggregation module, wherein each node in the GNN is the intermediate feature representation of an autonomous vehicle in the scene. Every agent's feature will be first transformed into others' coordinate systems and concatenate together on the channel dimension. Afterward, a simple 2D CNN is applied to the stacked feature to perform joint reasoning. DiscoNet [4] implements a convolutional graph neural network to perform multi-agent aggregations. It is different from V2VNet in 3 major aspects: 1) Instead of using CNN to directly fuse the stacked features, it utilizes CNN to calculate a coefficient matrix $N\times H\times W$ for all agents, where $N$ is the number of agents, $H$ and $W$ are the spatial resolutions of the BEV features. Then the final feature will be a weighted sum from all agents using the computed coefficient matrix. 2) Its CNN has more convolution and activation layers. 3) It also leverages a teacher model employing point cloud fusion to guide the training of the student model using intermediate fusion.
> - **Ablations of the proposed sparse attention module.**
>         We propose FAX attention for both multi-view camera fusion and multi-agent BEV fusion. Below we provide a stepwise ablation to validate the effectiveness of each component.
>   - To validate the efficacy of multi-view camera fusion for **single vehicle** application, we adopt the same backbone (EfficientNet) as CVT (CVPR2022) and only replace the Transformer in CVT with our proposed FAX Transformer, dubbed SinBEVT.
>         The below results demonstrate a fair comparison between multiple BEV segmentation methods, wherein our proposed SinBEVT outperforms other models.
>
>     | Method  | IoU (nuScenes) | IoU (OPV2V-Camera) |
>     |---------|----------------|--------------------|
>     | FIERY   | 35.8           | 37.1               |
>     | CVT     | 36.0           | 37.7               |
>     | SinBEVT | **37.1**           | **38.8**               |
>   - To validate only the performance of the multi-agent fusion, we adopt CVT as the backbone \textbf{for all the evaluated methods} and compare our multi-agent fusion model (FuseBEVT) with previous fusion methods in Table 1 in the main paper. We also experiment on the LiDAR data using the PointPillar backbone for a fair comparison in Table 2. Below we summarize the comparisons between our FuseBEVT and the state-of-the-art fusion methods V2VNet and DiscoNet **under the same backbone setting**.
>
>     | Method   | IoU (OPV2V-Camera) | AP@0.7 (OPV2V-LiDAR) |
>     |----------|--------------------|----------------------|
>     | V2VNet   | 53.5               | 82.2                 |
>     | DiscoNet | 52.9               | 83.6                 |
>     | FuseBEVT | **59.0**           | **85.2**             |
>   - Finally, we build the holistic CoBEVT model using the FAX attention for both multi-view camera fusion and multi-agent BEV fusion. The results below show a step-by-step performance improvement by building up CoBEVT from the vanilla no fusion CVT baseline.
>
>     | Method                    | IoU (OPV2V-Camera) |
>     |---------------------------|--------------------|
>     | CVT (baseline, no fusion) | 37.7               |
>     | SinBEVT (no fusion)       | 38.8 (+1.1)        |
>     | FuseBEVT (CVT + fusion)   | 59.0 (+20.2)       |
>     | CoBEVT (SinBEVT+FuseBEVT) | 60.4 (+1.4)        |

---

> > ### Author Response · Authors · 2022-08-22
> > **Response to Reviewer KVMN (cont.)**
> >
> > - **Regarding the comparison to the recent Arxiv paper for BEV segmentation ([https://arxiv.org/pdf/2206.07959.pdf](https://arxiv.org/pdf/2206.07959.pdf))**.
> >  We will also add the following discussions to the revised manuscript.
> >   - This Arxiv paper was published after our submission time on June 15, which by no means can we notice or compare against.
> >   - The Arxiv paper utilizes ResNet-101 as the backbone to extract features, which leads to **47.2M** parameter count and **7.3 FPS** on Tesla V100. On the contrary, our SinBEVT uses a much smaller backbone, EfficientNetV2, to extract features, which only has **1.6M** parameters with **51.2 FPS** on RTX3090 and 35 FPS on RTX2080. Our model is much more lightweight and designed for real-time applications.
> >     Therefore, it is unfair to directly compare the accuracy since our model uses a lightweight backbone and is around **30 times smaller**. There is another piece of evidence that can prove the choice of CNN backbone can largely influence the performance: Lift-Splat [6] achieves an IoU of 32.06 using ResNet-18 in the original paper, and that number rises to 42.1 when using ResNet-101 in ECCV2022 BEVFormer's experiments.
> >   - The Arxiv paper utilizes a larger batch size (i.e., 40 vs. 16) and higher RGB resolution (i.e., $448 \times 960$ vs. $224 \times 480$) compared to our CoBEVT. The Arxiv paper has demonstrated that the performance can drop by 3.3\% and 5.7\% if the batch size and image resolution decrease to 20 and $224 \times 480$. Therefore, it is unfair to directly compare the accuracy without having the same hyperparameter setting and training pipeline.
> >   - The Arxiv paper only validates on nuScenes without collaboration, which is a different task as we mainly study in this paper. It is unfair to compare performance across different datasets~(i.e. nuScenes vs. OPV2V). More importantly, we achieve 60.4 IoU on OPV2V with collaboration, higher than any single-vehicle baseline.
> >   - Our nuScenes experiments include all the lightweight models that have real-time performance. We compare our SinBEVT against a very recent model CVT (CVPR 2022), which we deem a strong and reliable benchmark.
> >
> > - **Hyperparameters such as resolution or batchsize.**
> >   - We follow prior works and use the same hyperparameters for a fair comparison.
> >     For example, all the multi-agent algorithms in OPV2V follow the same experimental setting~(e.g., batch size, learning rate, training epochs).
> >     Similarly, we use exactly the same hyperparameters and training schemes as CVT to validate our SinBEVT on the nuScenes dataset.
> >     Given this, our experiments can demonstrate the performance gains benefited only from our model designs regardless of hyperparameter effects.
> >   - Still, as suggested by the reviewer, we investigated the effects of hyperparameters including input resolution and batchsize using SinBEVT on the nuScenes dataset.
> >     As shown in the table below, increasing the input image resolution  can indeed bring performance gains, yielding a 39.3 IoU at $448\times 960$.
> >     However, unlike the Arxiv paper, our method is insensitive to the batch size.
> >
> >     | Method  | RGB resolution | Batch size | IOU  |
> >     |---------|----------------|------------|------|
> >     | SinBEVT | 224 x 480      | 8          | 36.6 |
> >     | SinBEVT | 224 x 480      | 16         | 37.1 |
> >     | SinBEVT | 224 x 480      | 32         | 37.0 |
> >     | SinBEVT | 112 x 240      | 16         | 32.4 |
> >     | SinBEVT | 448 x 960      | 16         | 39.3 |

---

> > > ### Author Response · Authors · 2022-08-22
> > > **Response to Reviewer KVMN (cont.)**
> > >
> > > - **Directly broadcast multi-view images vs. BEV features.**
> > > Broadcasting BEV features is a more practical solution than sending original multi-view camera images for three major reasons:
> > >   - **Scalability.** Modern autonomous vehicles usually have multiple cameras (e.g., nuScenes has 6 cameras, Waymo's vehicles have 29 cameras [8]), and sending all the images to one vehicle to perform detection/segmentation will cause a very high computation cost, thus unscalable. Assuming there are five agents in the collaboration, each with four cameras, we calculate the computation cost for these two different broadcasting approaches in the following table on an RTX3090. Note that the calculation of GFlops and FPS counts on the whole pipeline. It is clear to see that our approach has much fewer Flops and a faster inference speed. More importantly, when we increase the agent number to 6, the raw image collaboration strategy will exceed the GPU memory limitation (24GB). However, our approach can still reach 25 FPS even with 10 agents without exceeding the GPU memory cap.
> > >
> > >     | Broadcasting approach             | GFlops | FPS  |
> > >     |-----------------------------------|--------|------|
> > >     | Multi-view images (20 images)     | 1400   | 12.3 |
> > >     | BEV features (5 BEV feature maps) | 215    | 30.1 |
> > >   - **Bandwidth.** V2V communication has limited bandwidth, which results in a strict requirement on the transmission data size. Assume every vehicle has 4 cameras and each has a resolution of $800\times600$ (i.e., OPV2V's setup setting), then the transmission size is 23.04MB. This is not practical to deliver as the normal transmission rate is
> > >     3.75 MB/s [5]. The standard compression rate for images that avoids severe quality degradation is around $10\times$. Therefore, even with some compression, the data size is still around 2MB, which will cause severe communication delay. Furthermore, when the number of cameras increases (e.g., 8 cameras in Tesla), the bandwidth requirement will be linearly raised. On the contrary, our BEV features have only 512KB without any compression, and they can be compressed to only 33KB with a slight performance drop. More importantly, unlike delivering raw images, the bandwidth requirement for our approach is independent of the number of cameras.
> > >   - **Learning efficiency.** Many SOTA BEV segmentation methods (e.g., CVT, PETR [7]) utilize the camera extrinsics implicitly by applying an MLP to convert them into a D-dimensional positional embedding. This method is effective as the camera extrinsics are usually identical across different scenarios  because of the fixed mounted camera positions. However, for the camera images received from other vehicles, the extrinsics need to multiply with an external transformation matrix between ego vehicle and other agents. Since the relative position between two vehicles can be very random, this will make the final extrinsics similar to a noise matrix, and this will be challenging to let the MLP learn a useful positional embedding from the extrinsics.

---

> > > > ### Author Response · Authors · 2022-08-22
> > > > **Response to Reviewer KVMN (cont.)**
> > > >
> > > > - **Compare FAX with a generic 2D transformer in the BEV space.** We thank the reviewer for the great question. We have conducted additional ablation studies to compare against both 2D CNNs and transformers in the BEV space. Specifically, DiscoNet [4] utilizes a 2D CNN to fuse features, and we select it as the 2D CNN baseline. The original Swin Transformer cannot be directly applied to the stacked BEV features (of dimensions $(B,N,H,W,C)$, where $B$ is the batch size, $N$ denotes the number of agents, and $H,W,C$ denote the feature shape). Thus, we built two forms of Swin baselines: 1) **CatSwin**: stack the BEV features along the channel dimension as $(B,H,W,N\times C)$, then apply the vanilla 2D Swin Transformer; 2) **CoSwin**: replace the global attention in our FAX module with a shifted window attention. So the CoSwin contains a fused local and shifted local attention, similar to the original designs in Swin Transformer. We include a complete comparison with different numbers of layers as below. It can be observed that our CoBEVT outperforms all other methods in terms of various numbers of layers with even fewer parameters and low Flops. This demonstrates the excellent performance of our proposed architecture design.
> > > >
> > > >     | Method   | # of Layer | Params | GFlops | IoU      |
> > > >     |----------|------------|--------|--------|----------|
> > > >     | DiscoNet | 1          | 0.61   | 2.63   | 49.2     |
> > > >     | DiscoNet | 2          | 1.22   | 5.23   | 50.1     |
> > > >     | DiscoNet | 3          | 1.83   | 7.83   | 52.9     |
> > > >     | CatSwin  | 1          | 2.07   | 4.21   | 52.8     |
> > > >     | CatSwin  | 2          | 4.05   | 8.25   | 53.2     |
> > > >     | CatSwin  | 3          | 6.03   | 12.29  | 57.4     |
> > > >     | CoSwin   | 1          | 0.29   | 2.74   | 55.0     |
> > > >     | CoSwin   | 2          | 0.58   | 5.45   | 57.6     |
> > > >     | CoSwin   | 3          | 0.86   | 8.15   | 58.6     |
> > > >     | CoBEVT   | 1          | 0.29   | 2.75   | **57.3** |
> > > >     | CoBEVT   | 2          | 0.58   | 5.46   | **58.2** |
> > > >     | CoBEVT   | 3          | 0.86   | 8.17   | **60.4** |
> > > >
> > > > Best,
> > > >
> > > > Authors of Paper 142
> > > >
> > > > [1] Q. Chen, X. Ma, S. Tang, J. Guo, Q. Yang, and S. Fu. F-cooper: Feature based cooperative perception for autonomous vehicle edge computing system using 3d point clouds. In Proceedings of the 4th ACM/IEEE Symposium on Edge Computing, pages 88–100, 2019.
> > > >
> > > > [2] R. Xu, H. Xiang, X. Xia, X. Han, J. Liu, and J. Ma. Opv2v: An open benchmark dataset and fusion pipeline for perception with vehicle-to-vehicle communication. 2022 International Conference on Robotics and Automation (ICRA)
> > > >
> > > > [3] T.-H. Wang, S. Manivasagam, M. Liang, B. Yang, W. Zeng, and R. Urtasun. V2vnet: Vehicleto-vehicle communication for joint perception and prediction. In European Conference on Computer Vision, pages 605–621. Springer, 2020.
> > > >
> > > > [4] Y. Li, S. Ren, P. Wu, S. Chen, C. Feng, and W. Zhang. Learning distilled collaboration graph for multi-agent perception. Advances in Neural Information Processing Systems, 34, 2021.
> > > >
> > > > [5] Arena, F., Pau, G.: An overview of vehicular communications. Future Internet 11(2), 27 (2019) 12
> > > >
> > > > [6] Philion, J., Fidler, S.: Lift, splat, shoot: Encoding images from arbitrary camera rigs by implicitly unprojecting to 3d. In: European Conference on Computer Vision. pp. 194–210. Springer (2020)
> > > >
> > > > [7] Yingfei Liu, Tiancai Wang, Xiangyu Zhang, Jian Sun, PETR: Position Embedding Transformation for Multi-View 3D Object Detection, ECCV 2022
> > > >
> > > > [8] Liu, Zhijian, et al. "BEVFusion: Multi-Task Multi-Sensor Fusion with Unified Bird's-Eye View Representation." arXiv 2022.

---

### Official Review · Reviewer_dUwG · 2022-08-03

**Originality:** Good
**Technical Quality:** Good
**Clarity Of Presentation:** Fair
**Impact:** 3

**Recommendation:**

Weak Reject: I recommend rejecting the paper, but will not argue for my recommendation if the majority of other reviewers have a different opinion.

**Summary:**

The paper proposes CoBEVT, the first generic multi-agent multi-camera perception framework that can cooperatively generate BEV map predictions. Each view is first passed through an image encoding to extract image images, which are then in turn used to form BEV features. The features are concatenated and fused together to make BEV predictions. The method is tested on OPV2V and nuScenes.

**Issues:**

Similar to the weaknesses above.

**Quality Of The Limitations Section:**

Additional details required

**Reviewer Expertise:**

4: The reviewer is confident but not absolutely certain that the evaluation is correct

**Robotics Focus:**

Highly relevant to robotics but no hardware experiments

**Strengths And Weaknesses:**

Strengths

- The papers utilize the latest advance in transformer architectures to tackle the V2V communication problem.

Weaknesses

- It is not clear whether those proposed transformer structures are necessary for the problem setup or not. Although it makes sense, but there is no comparison with the proposed structure with the existing methods. For example, we can compare the BEV features produced by single camera with the existing models through map segmentation and 3D object detection on nuScenes to justify the model designs to extract features from single cameras.

- The experiment detail on nuScenses is really sparse, which leads to confusion. On Line 242, it claims that "Compared to the stat-of-the-art method CVT, we are 1.1% higher with similar parameters and latency." This statement is too vague, especially compared to the results on OPV2V. The supplementary material only provides quantitative results.

- It is strange to use ResNet34 as the image encoder while the authors try to put latest transformer models on top of it, instead of designing an end-to-end transformer for image encoding to BEV features.

- It is better to know where the multi-agent fusion can improve the prediction accuracies, and whether the improvements depend on the distances of the vehicles.

**Summary Of Recommendation:**

I am on the borderline for the paper, but leaning towards rejection now. While fusing multiple cameras for 3D predictions is an interesting topic, the present paper leaves too many details out to understand and justify the model designs. The experiment detail on nuScenes is too little for a comprehensive understanding.

---

> ### Author Response · Authors · 2022-08-22
> **Response to Reviewer dUwG**
>
> We appreciate the constructive comments by the reviewer. We address the questions raised by the reviewer below and hope it will help the reviewer to finalize the judgment.
>
>
> - **Whether those proposed transformer structures are necessary for the problem setup.**
>      Our paper aims to propose a framework that can effectively reason the correlations between multi-view camera images or multi-agent BEV features. Transformer, which utilizes the self-attention mechanisms to aggregate features dynamically and globally, is a natural fit for our use cases. However, vanilla transformers usually have a quadratic computation cost, which  cannot be deployed on real vehicles, especially for high-dimensional data like multi-view cameras or multi-agent features. To this end, we designed a 3D sparse attention module (FAX) for both multi-view and multi-agent fusion, which can enjoy the benefits of dynamic global attention while avoiding expensive computation. We have empirically demonstrated the superior performance of our proposed model over existing methods on multiple tasks, which can be found in Tables 1-3 in the paper.
>
> - **Comparison with the proposed structure with the existing methods for a single vehicle.** In the nuScenes dataset, we compare several SOTA methods, including CVT (CVPR2022) [1], Fiery (CVPR2021) [2], Lift-Splat (ECCV2020) [3], and VPN (RAL2020) [4] on BEV semantic segmentation. We demonstrate that our proposed single-vehicle multi-view camera fusion Transformer can outperform those methods while keeping lightweight (1.6M params) and fast (35FPS on RTX2080 and 51FPS on RTX3090).
>
>
> - **Experiment details on nuScenes.**
>     Thanks for bringing out this issue. For better understanding, we have added more experimental details of nuScenes in the supplementary file. We are also presenting the added details here. To make a fair comparison, we strictly follow the same experiment setting as CVT.
>
>   - **Image Encoder.** We follow CVT and Fiery to use EfficientNet B-4 as THE image feature extractor. We compute features at three scales - (56, 120), (28, 60), and (14, 30).
>
>   - **SinBEVT.** The BEV query starts with a size of $100\times100\times32$ and ends with a size of $25\times 25 \times 128$. We set the window/grid size of image features and BEV query for the three FAX-CA blocks as (6, 12), (6, 12), (14, 30), and (10, 10), (10, 10), (25, 25) respectively. Please refer to the  SinBEVT specifications shown in Table A2 in the supplementary file.
>
>   - **Decoder** The decoder structure is the same as CVT. The decoder consists of three (bilinear upsample + conv) layers to upsample the BEV feature to the final output size ($200\times200$).
>
>   - **Training.**  We train our models with focal loss and a batch size of 4 per GPU for 30 epochs. We employ  AdamW optimizer with the one-cycle learning rate scheduler. The whole training process is around 8 hours on 4 RTX3090 GPUs.
>
>   - **Evaluation.** We evaluate the 100m×100m area around the vehicle with a 50cm sampling resolution. We use the Intersection-over-Union (IoU) score between the model predictions and the ground-truth segmentation mask.
>
>
> - **ResNet34 as image encoder.**
>     The training dataset sizes for autonomous vehicles are relatively small, which are challenging to train a pure vision Transformer from scratch. Thus, most of the recent Transformer models for BEV understanding take advantage of a CNN encoder to extract camera features, then apply a Transformer model on top. E.g., CVT uses EfficienNet, DETR3D [5] and BEVFormer [6] use ResNet101, "Translating Image into Maps" (ICRA2022) [7] uses ResNet-50. Similarly, we follow these state-of-the-art works and use EfficientNet and ResNet for OPV2V and nuScenes.
> - **Prediction accuracy w.r.t. distance of vehicles.**
>     We have conducted extra experiments to calculate the accuracy of the vehicle predictions concerning different distance ranges.
>     For a given distance range (e.g., $<$20m), we masked out vehicles out of this range (e.g., $>$20m) both on groundtruth and prediction segmentation mask.
>     As shown in the table below, both SinBEVT and CoBEVT see performance drop with increased distance of the vehicles.
>     However, benefiting from V2V communication, CoBEVT demonstrates stronger robustness against distant vehicles, still achieving 42.5 IoU at $>$40m range.
>
> | Method  | <20m | 20-40m | >40m |
> |---------|------|--------|------|
> | SinBEVT | 64.9 | 28.4   | 2.93 |
> | CoBEVT  | 72.7 | 56.5   | 42.5 |
>
> Best,
>
> Authors of paper 142.

---

> > ### Author Response · Authors · 2022-08-22
> > **Response to Reviewer dUwG (cont.)**
> >
> > Here we attach the references for the response as the previous one exceeds the maximum length.
> >
> > [1] Zhou, Brady, and Philipp Krähenbühl. "Cross-view Transformers for real-time Map-view Semantic Segmentation." CVPR 2022.
> >
> > [2] Hu, Anthony, et al. FIERY: Future Instance Prediction in Bird's-Eye View From Surround Monocular Cameras." ICCV 2021.
> >
> > [3] Philion, Jonah, and Sanja Fidler. "Lift, splat, shoot: Encoding images from arbitrary camera rigs by implicitly unprojecting to 3d." ECCV 2020.
> >
> > [4] Pan, Bowen, et al. "Cross-view semantic segmentation for sensing surroundings." RAL 2020.
> >
> > [5] Wang, Yue, et al. "Detr3d: 3d object detection from multi-view images via 3d-to-2d queries." CoRL 2022.
> >
> > [6] Li, Zhiqi, et al. "BEVFormer: Learning Bird's-Eye-View Representation from Multi-Camera Images via Spatiotemporal Transformers." ECCV 2022.
> >
> > [7] Saha, A., Mendez, O., Russell, C., & Bowden, R.  "Translating images into maps." ICRA 2022.

---

> ### Author Response · Authors · 2022-08-22
> **Revised Supplementary Materials**
>
> We have attached the revised supplementary materials to this response. We have highlighted the changes in blue color.

---

### Official Review · Reviewer_Kibb · 2022-08-06

**Originality:** Very Good
**Technical Quality:** Very Good
**Clarity Of Presentation:** Very Good
**Impact:** 4

**Recommendation:**

Weak Accept: I recommend accepting the paper, but will not argue for my recommendation if the majority of other reviewers have a different opinion.

**Summary:**

The authors propose CoBEVT for bird's eye view (BEV) semantic segmentation from multiple cameras in a multi-agent environment in this paper. CoBERT is a Transformer-based method, in which the BEV representation is extracted by the SinBEVT Trasnformer in each vehicle, and then aggregated by FuseBEVT. These Transformers include a fused axial (FAX) module with both global and local attention, which reduces the computational cost and shows high performance in this problem setting. The proposed method has shown high performance in a variety of different settings at various metrics.

**Issues:**

As pointed out in "weakness", I would like the authors to discuss more how well the hypotheses in this setting are valid in the real world and how well the proposed method can be used in the real world.

**Quality Of The Limitations Section:**

Limitations are addressed clearly

**Reviewer Expertise:**

3: The reviewer is fairly confident that the evaluation is correct

**Robotics Focus:**

Relevant but unlikely to deploy to hardware in near future

**Strengths And Weaknesses:**

- Strengths:
  - The paper is well organized and readable.
  - The proposed method, FAX, is less expensive than full attention, yet very effective in the current problem setting, and could be useful in this research area in the future.
  - The experiments in this paper have been evaluated with various metrics (e.g., segmentation and processing speed) and settings (e.g., number of agents, camera dropout) on different methods and datasets, which are highly reliable.

- Weakness:
  - The authors assume that "poses of all the agents are accurate and transmitted messages are synchronized" but as the authors state in the limitation section, this is not necessarily the case in a real vehicle. Since this conference is also focused on the feasibility of hardware robots, the authors need more discussion on the extent to which these assumptions may not hold in reality, the extent to which these difficulties affect the current problem setting, and whether the proposed method has the potential to address these difficulties.

**Summary Of Recommendation:**

Although I am not an expert in BEV map understanding, I believe that the proposed method is novel and will be useful in this community in the future. I would like to accept this paper, although I have some concerns that it has not been sufficiently discussed for real-world applications.

---

> ### Author Response · Authors · 2022-08-22
> **Response to Reviewer Kibb**
>
> We thank the reviewer’s detailed comments and recognition of our novelty and state-of-the-art performance. Below are our responses to the points raised by the reviewer. We hope these will help improve the clarity of the paper.
>
>
> - **The authors assume that ''poses of all the agents are accurate and transmitted messages are synchronized'' but this is not necessarily the case in a real vehicle.**
>     We agree that our current assumption is not necessarily the case in a real vehicle.
>     However, this is our first attempt to validate the idea of the multi-agent multi-camera perception problem for autonomous driving.
>     And we have proposed a holistic Transformer architecture to tackle this problem, which has shown superior performance on both simulated and real datasets.
>     Our next step will involve communication delay and GPS error to simulate real-world scenarios better.
>
> - **Whether the proposed method has the potential to address these difficulties.**
>     Here we conduct an additional experiment to validate the performances under communication and GPS noise. We add a constant 100 ms communication delay by assigning one frame from the previous timestamp for all other agents except the ego vehicle. We further add GPS noise to the poses. The positional and heading noises of the transmitter are drawn from a Gaussian distribution with a default standard deviation of 0.5m and 0.5&deg; respectively. Although our proposed CoBEVT architecture does not explicitly account for these difficulties regarding communication noise and GPS error,
>     as shown in the table below, our model still performs the best with the smallest performance drop. We achieve a leading IoU of 55.0, which is still much better than the single vehicle(no fusion) result.
>
>
> | Method     | IoU(Perfect) | IoU(Noise) |
> |------------|--------------|------------|
> | No Fusion  | 37.7         | 37.7       |
> | Map Fusion | 45.1         | 39.1       |
> | AttFuse    | 51.9         | 44.2       |
> | F-Cooper   | 52.5         | 44.1       |
> | V2VNet     | 53.5         | 47.1       |
> | DiscoNet   | 52.9         | 46.4       |
> | CoBEVT     | **60.4**     | **55.0**   |
>
> Best,
>
> Authors of Paper 142

---

### Author Response · Authors · 2022-08-22
**Revised Main Paper(pdf) and Supplementary Materials(zip)**

**Comment:**

We have attached the paper and supplementary materials with revisions. All changes are highlighted in blue color.

**Zip File:**

/attachment/81498b65e2994a847d758efcd6bbd530d7260101.zip

---

### Author Response · Authors · 2022-08-25
**Sincerely expecting further discussions**

Thank you for the constructive comments and feedback.  We hope our response can help address your concerns and better clarify our paper. We would be more than happy to provide further clarification if more discussions are needed. Hope our response can lead to a fair and positive assessment.

---

### Meta-Review · Area_Chair_cELw · 2022-08-15

**Recommendation:** Accept (Poster)
**Confidence:** 4

**Metareview:**

The paper is well-written and well-organized. It presents an interesting and novel method, especially with regard to the fused axial attention model, which could potentially have a major impact. While some of the reviewers were convinced by the experimental evaluation, others pointed to important shortcomings:

* motivation of the approach should be made clearer: which problems do the transformers actually solve in this setting?
* more details should be discussed on where multi-agent fusion can improve the prediction accuracies, and whether the improvements depend on the distances of the vehicles.
* notation (especially in section 3.1) should be revised to make the difference between local and global attention more clear
* the paper contains a significant number of typos and grammatical mistakes which should be corrected
* at least for BEV segmentation, other approaches in the literature exist which achieve better performance without transformers for fusion of information and without multi-agent collaboration, so are the conclusions of the paper really justified?
* evaluation is insufficient to really assess progress: sparse transformer should be evaluated against other natural candidates in BEV space, such as generic 2D transformers, SWIN transformers or a 2D CNN
* robustness to hyperparameter changes is not investigated
* the authors assume an unrealistic setting with accurate and synchronized transmission of agent poses, and the potential of the method to overcome this limitation should be discussed

===

Post rebuttal/discussion update:

The reviewers appreciated the extensive effort which the authors put into their rebuttal. Although weaknesses with respect to clarity, presentation, and explanation of the design choices remain (and should be improved), overall, the rebuttal addressed concerns in a rather comprehensive manner.  The method is interesting, novel and offers promising empirical results.

---

> ### Author Response · Authors · 2022-08-22
> **Response to Meta Reviewer**
>
> We thank all reviewers' constructive comments. We appreciate the recognition of the novelty of our method and the importance of the problem that we try to address by AC and other reviewers. We are responding below to the concerns mentioned by the AC and summarize the major points of our response. We have also uploaded a revised version of the paper and supplementary file.
>
> - **Motivation of the approach**. We mainly study the multi-view-camera multi-agent  BEV map segmentation problem. This setting requires a framework to effectively reason the correlations between multi-view camera images and multi-agent BEV features in real-time speed. To meet this demand, we designed a 3D sparse attention module (FAX) for both multi-view and multi-agent fusion, which enjoy the benefits of dynamic and global attention brought by the Transformer architecture while avoiding the expensive computation that standard Transformers have.
> - **More details on multi-agent fusion and distance.**     Our proposed multi-agent fusion can better handle the prediction accuracies on both occluded and distant vehicles. We have highlighted those areas for better visualization in Fig. 4 of the main paper and Figs. 4 and 5 in the Appendix.
>     We also provide additional experimental results with respect to distance in **Response to Reviewer dUwG**. The results show that our multi-agent fusion can improve the performance of the single-agent at all distance ranges, and the boost is most significant at the far distance.
> - **Notation.**  Our notation is very close to our implementations, which we feel can help readers better parse out code. To help for better understanding, we have added an entire section (Sec. 2) in the Appendix to provide all the details on the proposed FAX module.
> - **Typos and grammatical mistakes.**  We thank the reviewers for all the corrections. We have fixed all the issues the reviewers spotted and proofread again to further improve the readability.
> - **Other BEV segmentation approaches.**     We have provided a comprehensive reply regarding the comparison to the recent Arxiv paper that reviewer KVMN mentioned for BEV segmentation in our **Response to Reviewer KVMN**.
>     We are also summarizing the major points here:
>    1. The Arxiv paper was published after our submission time.
>    2. The Arxiv paper uses a much heavier image feature extraction backbone than ours, which results in a 30 times larger model size, so it is not directly comparable. Our training pipeline and hyperparameters are totally different, e.g., we use smaller input resolution and batch size. Thus, it is unfair to directly compare the numbers. Furthermore, our model can achieve real-time performance (51FPS on RTX3090) while the Arxiv paper is only 7FPS on Tesla V100, which shouldn't be ignored. Our nuScenes experiments include all the lightweight models with real-time performance, including a very recent model, CVT (CVPR2022), which we deem a strong and reliable benchmark.
>    3. We have demonstrated that, with V2V collaboration, our model can reach an IoU of 60.4 on the OPV2V dataset, which is higher than any single-vehicle performance.
>
> - **Evaluation against Swin and CNN.** In **Response to Reviewer KVMN**, we have presented additional comparisons against other neural candidates in BEV space, including variants of Swin Transformers and 2D CNN. The results show that our design yield the best performance with very few parameters and FLOPs.
> - **Robustness to hyperparameters.**  In **Response to Reviewer KVMN**, we have provided additional ablation studies on the effects of hyperparameter changes, including input resolution and batch size.
> - **Assumed setting.** We have provided our response together with the additional experimental results under noise settings in **Response to Reviewer Kibb**.
>
> Given the significance of the challenging camera-based cooperative perception problem, the technical novelties/contributions, and the validated experimental excellence of the proposed CoBEVT method, we genuinely hope the discussions above and below could lead to a fair assessment of the proposed method -- the first attempt to tackle the V2V BEV segmentation problem.
>
> Sincerely,
>
> Authors of paper 142